# Accumulation of Health-Promoting Compounds in Upland Black Rice by Interacting Mycorrhizal and Endophytic Fungi

**DOI:** 10.3390/jof9121152

**Published:** 2023-11-29

**Authors:** Sabaiporn Nacoon, Wasan Seemakram, Thanawan Gateta, Piyada Theerakulpisut, Jirawat Sanitchon, Thomas W. Kuyper, Sophon Boonlue

**Affiliations:** 1Department of Microbiology, Faculty of Science, Khon Kaen University, Khon Kaen 40002, Thailand; n_sabaiporn@kkumail.com (S.N.); seemakram.w@kkumail.com (W.S.); thanawan.gateta@kkumail.com (T.G.); 2Department of Biology, Faculty of Science, Khon Kaen University, Khon Kaen 40002, Thailand; piythe@kku.ac.th; 3Salt-Tolerant Rice Research Group, Khon Kaen University, Khon Kaen 40002, Thailand; 4Department of Agronomy, Faculty of Agriculture, Khon Kaen University, Khon Kaen 40002, Thailand; jirawat@kku.ac.th; 5Soil Biology Group, Wageningen University & Research, P.O. Box 47, 6700 AA Wageningen, The Netherlands; thom.kuyper@wur.nl

**Keywords:** *Trichoderma zelobreve*, *Rhizophagus variabilis*, Maled Phai, black rice

## Abstract

There is an increasing interest in finding eco-friendly and safe approaches to increase agricultural productivity and deliver healthy foods. Arbuscular mycorrhizal fungi (AMF) and endophytic fungi (EPF) are important components of sustainable agriculture in view of their ability to increase productivity and various plant secondary metabolites with health-promoting effects. In a pot experiment, our main research question was to evaluate the additive and synergistic effects of an AMF and four root-endophytic fungi on plant performance and on the accumulation of health-promoting secondary compounds. Plant growth varied between the treatments with both single inoculants and co-inoculation of an AMF and four EPF strains. We found that inoculation with a single EPF positively affected the growth and biomass production of most of the plant-endophyte consortia examined. The introduction of AMF into this experiment (dual inoculation) had a beneficial effect on plant growth and yield. AMF, *Rhizophagus variabilis* KS-02 co-inoculated with EPF, *Trichoderma zelobreve* PBMP16 increased the highest biomass, exceeding the growth rate of non-inoculated plants. Co-inoculated *R. variabilis* KS-02 and *T. zelobreve* PBMP16 had significantly greater beneficial effects on almost all aspects of plant growth, photosynthesis-related parameters, and yield. It also promoted root growth quality and plant nutrient uptake. The phenolic compounds, anthocyanin, and antioxidant capacity in rice seeds harvested from plants co-inoculated with AMF and EPF were dramatically increased compared with those from non-inoculated plants. In conclusion, our results indicated that EPF and AMF contributed to symbiosis in Maled Phai cultivar and were coordinately involved in promoting plant growth performance under a pot trial.

## 1. Introduction

Black rice (also known as purple rice) constitutes a group of varieties of rice (*Oryza sativa* L.), recognizable by their dark seed color. Black rice is grown mainly in Asia, especially in India, China, Sri Lanka, Indonesia, the Philippines, Laos, and Thailand [1,2]. Many varieties of colored rice have been introduced in Thailand and these provide a range of desirable colors. They contain higher levels of anthocyanins and phenolic compounds than the well-known white rice varieties [3]. Maled Phai cultivar is a non-glutinous, pigmented, upland rice that originated from southern Thailand, characterized by brown-purple seed. Its cultivation period is from the beginning of June to the end of October. It has been developed at the Agronomy Station, Khon Kaen University, and is currently widely distributed to local communities because of its good flavor and health promoting effects [4]. 

Dietary supplementation with black rice has a great impact on human health [5] and this drives increased demand. However, escalating prices of mineral fertilizers limits increases in its production. Agronomists and soil scientists have therefore started to look for alternative ways to increase production by focusing on other sources of nutrients and/or by harnessing soil biological processes that increase the efficiency of nutrient acquisition and use. Khan et al. [6] referred to the importance of using available organic manures efficiently through suitable application methods and targeted time of application. They also referred to integrated nutrient management practices by combining mineral fertilizers with organic manure. Increasing rice yield by minimizing nutrient losses to the environment has also been considered [7,8]. Previous studies reported that mineral fertilization is a quick method of providing plants with their required nutrients; however, due to its potential adverse effects on the environment with non-judicious use, it cannot be considered as the sole suitable method for yield increases. Accordingly, harnessing the use of soil microbes has been proposed as an alternative method to mineral fertilization. Results have shown that this can be an equally effective method [8,9]. The use of plant growth-promoting microorganisms such as arbuscular mycorrhizal fungi (AMF) or root-endophytic fungi (EPF) alone or in combination has been reported to improve crop productivity, while simultaneously contributing to soil health and sustainability [10,11,12]. Therefore, it is imperative to compare the production efficiency of different rice production systems under mineral fertilizer and the application of microbial consortia.

AMF establish mutualistic associations with the roots of most terrestrial plants, including rice. Mycorrhizal plants have a greater ability to acquire nutrients and soil water than non-mycorrhizal plants, resulting in increased plant fitness both under normal and adverse environmental conditions [13]. Previous studies reported the beneficial effects of AMF on the growth promotion of several plants, including rice (*Oryza sativa*) [14,15,16,17,18,19]. Endophytic fungi (EPF) live inside plant tissues or organs such as seeds, leaves, flowers, twigs, stems, and roots, but do not form a symbiotic interface, contrary to AMF. They colonize healthy plant tissues without causing visible symptoms to their hostsa. EPF can enhance plant growth by increasing nutrient acquisition, reducing disease severity, improving host tolerance of adverse environmental conditions, and enhancing the capability of the host plant to produce phytohormones (gibberellins (GA_3_), auxins like indole-3-acetic acid (IAA), cytokinin, abscisic acid), and secondary metabolites (phenols and flavonoids) [20]. EPF are one of the best sources of natural bioactive compounds that have potential applications in different fields, such as agriculture, medicine, and the food industry [21]. Several studies have documented that EPF have plant growth-promoting effects in several crops, such as corn, rice, and rye [22], watermelon (*Citrullus lanatus* L.) [23], sunchoke (*Helianthus tuberosus* L.) [24,25], and wheat (*Triticum aestivum* L.) [26]. Moreover, several species of EPF produce plant growth hormones, especially IAA and GA_3_ [27]. The synergistic effects of leaf-inhabiting EPF of grasses and AMF have been reported by Green et al. [28]. However, studies on the interactions between AMF and root-inhabiting EPF are scarcer. Competition for root space could be important in such cases and this competition could limit synergistic effects or even cause antagonism or negative synergism, as has been reported for the combination of AMF and *Trichoderma harzianum* [29]. An earlier study on the isolation and characterization of the plant growth-promoting properties of EPF in upland black rice (Maled Phai cultivar) showed that EPF increased rice growth and induced higher anthocyanin and phenolic compounds in seeds [30]. However, co-inoculation of AMF and EPF has not been conducted on black rice. Considering increasing demands for black rice in global markets, it is important that the productivity of black rice per hectare increases. Currently, productivity of black rice is relatively low, and this constrains its profitability for resource-poor farmers. Productivity increases would be possible by applying mineral fertilizers, but these can be costly and hence reduce profitability. Harnessing beneficial soil biota such as AMF or endophytic fungi for productivity increases is likely a viable and sustainable alternative under such conditions. Therefore, this work investigated the potential of single and co-inoculation of EPF and AMF for enhancing the growth of black rice and their ability to increase the concentration of phytochemicals (anthocyanin and phenolic compounds) and antioxidant activity in rice seeds. 

## 2. Materials and Methods

### 2.1. Preparation of Rice Seedlings

Seeds of upland black rice (*Oryza sativa* L.) cultivar Maled Phai were obtained from the Rice Project, Agriculture Faculty, Khon Kaen University, Thailand. Rice seeds were sterilized by soaking in 6% sodium hypochlorite solution for 10 min, washing thoroughly with distilled water, and then the seeds were germinated on wet filter paper. They were then transferred to germination plug trays containing twice-sterilized soils. Rice seedlings were grown under a greenhouse at 30–35 °C conditions and irrigated daily with tap water for approximately 14 days. Then, the uniform rice seedlings were transplanted to the pot trial.

### 2.2. AMF Inoculum Preparation

The AMF used in this study were molecularly identified as *Rhizophagus variabilis* strain KS-02 [14]. They was obtained from the Mycorrhiza and Mycotechnology Laboratory, Department of Microbiology, Faculty of Science, Khon Kaen University, Thailand. These fungi strongly promoted the growth of Maled Phai [14]. AMF soil inoculum was multiplied by the pot culture technique, using maize (*Zea mays* L.) as a host plant. Briefly, the soil was twice sterilized by autoclaving at 121 °C for 2 h and then added to 20-cm-diameter pots. Maize seeds were surface-sterilized by soaking in 10% sodium hypochlorite solution for 30 min before adding them to the pots. Spores of AMF (*R. variabilis*) were added to the pots containing these maize seeds. Maize was grown in a greenhouse at 30–35 °C. After 90 days, the plants were allowed to dry out, which caused sporulation of the AMF. The above plants were cut off, the soil was air-dried, and then they were ground into fine particles (<0.2 mm). Dried soils containing AMF spores, mycelia, and colonized root fragments were then used as the soil inoculum in the pot trial. The AMF inoculum was applied touching the roots of 14-day-old rice seeds at a rate of approximately 200 spores per pot.

### 2.3. EPF Inoculum Preparation

Four fungal endophytes including *Trichoderma zelobreve* PBMP16, *Talaromyces pinophilus* PBMP28, *Aspergillus flavus* KKMP34, and *Trichoderma harzianum* PBMP43 were obtained from the Mycorrhiza and Mycotechnology Laboratory, Department of Microbiology, Faculty of Science, Khon Kaen University, Thailand. These species produce various plant growth-promoting (PGP) properties and promoted the growth performance of Maled Phai. These fungi were inoculated on potato dextrose agar (PDA) and incubated for 7 days. After incubation, a disc of mycelium tip was cut by a 0.5 mm diameter cork borer and transferred to a tube that contained 30 g of sterilized sorghum (*Sorghum bicolor* L.) seeds. The tube was incubated under static conditions until full colonization on sorghum seeds by all fungi [29]. Ten sorghum seeds containing EPF mycelium were added onto the 14-day-old rice seedlings adjacent to the roots.

### 2.4. Soil Preparation for Rice Cultivation

The soil was a sandy loam soil with a pH of 7.26, an electrical conductivity (EC) of 0.043 dS m^−1^, organic matter (OM) content of 6.4 g kg^−1^, available nitrogen content of 0.024 mg kg^−1^, total phosphorus content of 146 mg kg^−1^, available P content of 61.4 mg kg^−1^, total potassium content of 428 mg kg^−1^, exchangeable K content of 50.2 mg kg^−1^, total calcium content of 655 mg kg^−1^, and sodium content of 50 mg kg^−1^. Wood chips, rocks, and plant debris in a soil sample were removed. The soil samples were sterilized autoclaving at 121 °C for 2 h. The soils were left at room temperature overnight and then were sterilized again at the same conditions before being packed in 20-cm-diameter pots containing 5 kg of soil in each pot.

### 2.5. Experimental Design

The experiment was conducted in an enclosed greenhouse at Khon Kaen University, Khon Kaen, Thailand. The experiment was arranged as a Randomized Complete Block Design (RCBD). The experiment was a factorial experiment with two factors, AMF (presence or absence of *R. variabilis*) and EPF (each of the four species plus non-inoculated control). In order to compare the effect of soil fungi with regular agronomic practices, we added a further control with mineral fertilizer (NPK 15-15-15; at 0.32 g per pot). In all, there were 11 treatments with four replications. The experiment was carried out for 120 days, when plants were harvested. 

### 2.6. Determination of Plant Growth and Photosynthesis-Related Characters

Plant growth parameters were assessed at the harvesting stage. The first fully expanded leaves from the top of each plant were analyzed for net photosynthetic rate (P_n_), stomatal conductance (g_s_), and transpiration rate (T_r_) using an LI-6400XT portable photosynthesis system (Li-Cor Inc., Lincoln, NE, USA). The conditions during gas exchange measurements were controlled as follows: photosynthetically active radiation (PAR) at 1200 μmol photon m^−2^ s^−1^, CO_2_ concentration at 400 μmol mol^−1^, and temperature at 30 ± 2 °C. The water use efficiency (WUE) was calculated by dividing the P_n_ by T_r_. Fresh roots were collected and scanned using a Epson Perfection V800 Photo scanner (SEIKO EPSON CORP., Amsterdam, The Netherlands) and then measured for root length and root diameter using the WinRhizo Pro2004a software (REGENT Instruments Inc., Québec, QC, Canada). The dry weights of the rice seed, roots, stems, and leaves were determined after drying at 80 °C for 3 days. The shoot nitrogen (N) was measured using the Kjeldahl digestion method by flow injection analysis (FIA), phosphorus (P) was determined by its reaction with molybdate and crystal violet, which was then measured using the spectrophotometric method, and potassium (K) was determined by a flame photometer after wet digestion. 

The chlorophyll concentration in the plant was measured from 100 mg of fresh leaves soaked in 80% acetone solution (25 mL) in dark conditions at room temperature (28 ± 2 °C). The absorbance solution was measured at wavelengths of 645 and 663 nm using a spectrophotometer [31]. The chlorophyll concentration was expressed as mg g^−1^ shoot dry weight.

### 2.7. Assessment of AMF Spore and AMF, EPF Root Colonization

Root samples were carefully washed and cleared in 10% potassium hydroxide (KOH) at 90 °C for 30 min, rinsed in tap water, acidified in 1% hydrochloric acid solution (HCl) overnight, and stained with 0.05% trypan blue in lacto-glycerol, according to Koske & Gemma [32]. The AMF colonization was assessed using the method of Trouvelot et al. [33]. The EPF root colonization was determined in 30 fragments of roots from different zones per treatment. EPF hyphae were separated from the AMF by the presence of mycelium septa. The colonization frequency of EPF in plant roots was calculated according to Mehmood et al. [34]: % Colonization frequency = (Total number of root colonized × 100)/(Total number of roots)(1)

### 2.8. Phytochemical Analysis of Rice Seeds

The phytochemicals in the rice seed investigated included anthocyanin (TAC),phenolic compounds (TPC), and 1-diphenyl-2-picrylhydrazyl radical (DPPH). The seed extracts used for the determination of TAC, TPC, and DDPH were obtained according to the method described earlier with some modifications [35]. Briefly, the samples of dried seed were finely ground, and an amount of 1.0 g was extracted with 10 mL methanol, shaken for 2 h, and then centrifuged at 3000 rpm for 10 min. The mixture was filtered (Whatman No. 1 filter paper), and the residues were re-extracted twice with 5 mL methanol using the same procedure. The three aliquots were combined and stored at −40 °C in the dark until analysis. 

The total anthocyanin content (TAC) was measured according to Lee [36] with some modifications. Two dilutions of the sample extracts (50 µL) were prepared: one for pH 1.0 using 0.025 M potassium chloride (KCl) buffer and the other for pH 4.5 using sodium acetate buffer (0.4 M). The mixture was then allowed to stand for 20 min before measuring the absorbance at 520 and 700 nm. The TAC was calculated using the following equation and expressed as cyanidin-3-glucoside equivalent per 100 g sample: Total anthocyanins (mg/100 g) = ∆A × MW × D × (V/G)/(€ × L) × 100(2)
where ∆A is absorbance = (A520 nm − A700 nm) pH 1.0 − (A520 nm–A700 nm) pH 4.5, € is the molar extinction coefficient of Cy-3-G = 29,600 M^−1^ cm^−1^, L is the cell path length of the cuvette = 1 cm, MW is the molecular weight of anthocyanins = 449.2 g mol^−1^, D is the dilution factor, V is the final volume (mL), and G is the weight of sample (g).

The total phenolic content (TPC) was determined using the method of Dewanto et al. [37] based on 125 µL of extracted samples and 250 µL Folin–Ciocalteu’s reagent, followed by the addition of 3 mL distilled water. The solution was mixed well and then allowed to stand for 6 min, after which 2.5 mL of 7% sodium carbonate (Na_2_CO_3_) was added. The reaction mixture was allowed to stand for 90 min at room temperature before measuring the absorbance at 760 nm (Hitachi High-Tech Science Corporation, Tokyo, Japan). Gallic acid was used as a calibration standard, and the results were expressed as mg gallic acid equivalent per 100 g sample. 

The DPPH free radical scavenging activity was determined according to the method described by Leong & Shui [38] with some modifications. Briefly, freshly prepared 0.1 mM solution of DPPH in methanol was prepared with absorbance 517 nm. An aliquot of 100 µL of each seed-extracted sample was mixed with 4.0 mL of DPPH solution and then allowed to stand at room temperature for 30 min before measurement. The percentage of radical scavenging ability was calculated by using the formula:Scavenging ability (%) = (Absorbance 515 nm of control) − (Absorbance 515 nm of sample)/(Absorbance 515 nm of control) × 100(3)

### 2.9. Statistical Analysis

Data were analyzed by one-way and two-way analysis of variance (ANOVA). A two-way ANOVA was used to investigate the possible synergism between AMF and EPF. A one-way ANOVA was additionally applied to compare the soil biological treatments with the addition of mineral fertilizer. Data were tested for normality and homogeneity of variances, and ANOVA assumptions were applied in all cases. A least significant difference (LSD) test was applied to test for significant differences among the means of different treatments at *p*-value < 0.05. All statistical analyses were performed using the Statistix 10 software.

## 3. Results

The effects of AMF and EPF on root colonization are shown in Table 1. Non-mycorrhizal treatments (control; with different species of EPF) remained (almost) free of mycorrhizal colonization, whereas the treatment with *R. variabilis* alone resulted in 24% root colonization. Co-inoculation of AMF and EPF resulted in a weak (in the case of *T. zelobreve*) or very strong (in the case of the three other species) reduction in mycorrhizal colonization. Inoculation with EPF resulted in high frequencies for all four species, whereas the treatment with AMF and the non-mycorrhizal control with mineral fertilizer resulted in (very) low frequencies. The fungal species’ identity was not determined in these cases. Co-inoculation of AMF and EPF resulted in minor declines in root colonization by EPF by *T. pinophilus* and *A. flavus* and a considerable decline in colonization by *T. harzianum*. Co-inoculation of AMF and *T. zelobreve* had no effect on colonization by the latter species, confirming the compatibility of the combination. 

The effects of AMF, EPF, and their combinations on plant performance are shown in Table 2, Table 3 and Table 4. For the shoot biomass, EPF was a significant source of variation, whereas AMF and the interaction AMF × EPF were not. The shoot biomass was highest for plants inoculated with *T. zelobreve*, independent of the absence or presence of AMF. Without AMF, plants inoculated with *A. flavus* remained small, but in the combination AMF plus *A. flavus*, the shoot biomass was not different from the AMF plus EPF treatments, other than the combination of AMF plus *T. zelobreve*. The application of mineral fertilizer also significantly increased the shoot biomass compared with the unfertilized control. For root biomass, AMF, EPF, and the interaction AMF × EPF were significant sources of variation. The root biomass was higher for mycorrhizal plants than for non-mycorrhizal plants and within EPF plants inoculated with *T. zelobreve* tended to have higher root biomass than plants inoculated with the other species of EPF. The combination of AMF plus EPF had less root biomass than plants inoculated with AMF only, indicating negative synergy between both fungal groups on root performance. In addition, Table 2 shows the results of the root parameters. Mycorrhizal plants had a slightly, but significantly, smaller root diameter than non-mycorrhizal plants, both in the absence and presence of EPF. There was variation in root diameter between species of EPF, but there was no influence of AMF on differences in root diameter in plants colonized by different EPF. 

The grain yield was significantly affected by EPF and the interaction AMF × EPF. In the absence of EPF, the grain yield was significantly higher in mycorrhizal plants than in non-inoculated control plants, with an increase of more than 100%. Plants inoculated with *T. zelobreve* achieved higher yields than plants inoculated with the other species of EPF. The significant interaction term AMF × EPF was due to the significant yield reduction when mycorrhizal plants were inoculated with the other species of EPF and especially with *A. flavus*. The application of mineral fertilizer also increased the yield compared with the unfertilized control, but the effect was somewhat less than when plants were inoculated with *R. variabilis*, *T. zelobreve*, or the combination (Table 2).

The shoot nutrient content was significantly correlated with the shoot biomass. For all three nutrients, both AMF and EPF were significant sources of variation, whereas the interaction AMF × EPF was only significant for Mycorrhizal plants and had higher nutrient contents than non-mycorrhizal plants. Plants inoculated with both AMF and EPF had higher nutrient contents than plants only inoculated with EPF, and the lack of a significant interaction term indicated that the effects were additive. The highest shoot K content was observed in plants inoculated with *T. zelobreve*, both in the absence and presence of AMF. The application of mineral fertilizer generally resulted in lower N and P contents than when plants were inoculated with AMF or EPF, which was a difference hardly observed in the case of K (Table 3).

For chlorophyll concentration (chlorophyll a, chlorophyll b, and total chlorophyll, which were all significantly correlated), AMF, EPF and the interaction AMF × EPF were significant sources of variation. Mycorrhizal plants had higher chlorophyll concentrations than non-mycorrhizal plants. Plants inoculated with *T. zelobreve* had higher chlorophyll concentrations than plants inoculated with the three other species of EPF. The combination of AMF plus EPF resulted in higher chlorophyll concentrations than when EPF were inoculated in the absence of AMF; however, the chlorophyll concentrations remained lower than with plants only inoculated with AMF, still indicating some negative synergy. Plants that received mineral fertilizer had the highest concentration of chlorophyll, higher than in any inoculation treatment (Table 4).

For the rate of photosynthesis, both AMF and EPF were significant sources of variation. In the presence of AMF, photosynthesis was higher than in its absence. Within EPF, the photosynthesis rate was highest when inoculated with *T. zelobreve* and lowest when inoculated with *A. flavus*, both when non-mycorrhizal and when mycorrhizal. In the absence of EPF, AMF inoculation resulted in a significant increase in stomatal conductance, independent of whether plants received mineral fertilizer. Inoculation with EPF increased stomatal conductance in almost all cases, with the largest effect in plants that were inoculated with *T. zelobreve*, both when non-mycorrhizal and when mycorrhizal. The significant interaction term was caused by the negative synergy when plants were inoculated with AMF and *T. pinophilus*. Inoculation with EPF increased the transpiration rate and hence reduced the water use efficiency in three of the four species studied, both in the non-mycorrhizal and mycorrhizal condition. However, inoculation with *T. zelobreve* reduced transpiration and hence increased the water use efficiency, both when non-mycorrhizal and mycorrhizal (Table 4). 

Plants without both AMF and EPF had the lowest antioxidant activity. Compared to non-inoculated plants, plants that were solely colonized by AMF had higher antioxidant activity. Plants colonized by EPF had the highest activity and the presence of AMF slightly reduced that activity. The highest values for antioxidant activity were recorded for plants inoculated by *T. zelobreve*, both in the absence and presence of AMF (Figure 1). Plants without AMF and EPF had very low anthocyanin levels, and both classes of root colonizing fungi increased anthocyanin levels. In the absence of EPF, inoculation with AMF increased anthocyanin levels, but in the presence of EPF, plants inoculated with AMF showed reduced anthocyanin levels. Plants with EPF and without AMF had higher anthocyanin levels than plants with EPF and AMF. There was also large variation between the different species of EPF, also to the extent that the combination of EPF and AMF modified anthocyanin levels (Figure 1), and a clear pattern was not evident. The highest anthocyanin levels were recorded for plants inoculated with *T. zelobreve* in the non-mycorrhizal condition. The concentration of phenolics exhibited the same pattern, with EPF and the interaction EPF × AMF being significant sources of variation. In the absence of AMF and EPF, the application of mineral fertilizer was less effective than inoculation with either class of fungi (Figure 1). 

## 4. Discussion

The role of AMF in both increasing plant size and increasing concentrations of health-promoting secondary compounds has been well-known for a long time [39,40,41,42,43]. Such effects have also been reported for rice, particularly black rice. The upregulation of anthocyanin biosynthesis in black rice as a consequence of AMF symbiosis has also been reported by Tisarum et al. [44] and Wangiyana et al. [45]. A previous study on Maled Phai black rice cultivar has shown that AMF-inoculated plants exhibit higher productivity and concentrations of phytochemicals compared with non-mycorrhizal plants [14]. In the current study, the presence of AMF remarkably improved the root dry weight, grain weight, antioxidants, and anthocyanin, compared with non-inoculated plants. The application of mineral fertilizer in the absence of AMF, largely, but not completely, achieved the same enhanced plant performance (Table 2, Figure 1).

The role of EPF in enhancing plant performance is only more recently gathering momentum. Whereas the beneficial roles of endophytic bacteria have been widely reported (for rice: [46,47,48]), documenting the roles of EPF has lagged behind. EPF have the ability to colonize the host plant’s interior tissues and build a symbiotic association with their host plants. They do not form a symbiotic interface and for that reason cannot be considered equivalent to mycorrhizal fungi. EPF exhibit a range of relationships with their hosts including mutualism, commensalism, and parasitism [49,50]. The outcome of the interaction between a species of EPF and a plant species depends both on the nature of the fungal and plant species but also on environmental conditions. Some endophytes only exhibit a mutualistic interaction with one plant species but not with another [51]. The genotype of the host and fungus are important factors to determine the establishment of a symbiotic relation [52]. Under more favorable conditions, the outcome of the interaction is frequently neutral or negative, whereas under unfavorable environmental conditions (drought) the outcome is more often beneficial for the plant [53].

Studies on EPF had often concentrated on their roles in increasing the ability of plants to become better protected against pathogens [54,55] and (physiological) drought due to too little water or too much salt [56]. Recent studies have also provided evidence for their growth-enhancing effects on rice [44,57,58,59,60,61,62]. Previous research by Gateta et al. [30] reported the isolation of a total of four strains of endophytic fungi from upland black rice with respect to plant growth-promoting (PGP) properties. The four endophytic fungi possessed several relevant properties, including IAA production, siderophore production, ammonia production, inorganic phosphate solubilization, and gibberellic acid production. The result was similar to studies on EPF from wheat (*Triticum aestivum* L.) [63], lentil (*Lens culinaris* L.) [64], and the medicinal plant *Teucrium polium* L. [65]. In our study, all four species of EPF produced significantly more plant biomass and grain yield than the unfertilized, non-mycorrhizal control, indicating their potential to enhance plant performance. When grown alone, their beneficial effects were of a comparable magnitude as that of AMF; however, the combination of AMF with *T. pinophilus* or *A. flavus* significantly reduced these benefits, showing clear antagonism. We will therefore refrain from further discussing those species as application in an agricultural context, where AMF are always present, is very unlikely to be beneficial so we will focus on both species of *Trichoderma*.

The mechanisms of *Trichoderma* in promoting plant growth involve the production of auxin-like compounds, improving the availability of nutrients, affecting the root system, and inducing systemic resistance [66,67]. For example, strain T969 of *T. harzianum* increased the shoot height, shoot diameter, and root weight of tomato (*Solanum lycopersicum*) seedlings [68]. Inoculation with strain Snef1910 of *T. citrinoviride* resulted in increasing the plant growth of tomatoes in pot and field experiments [69]. Additionally, *T. citrinoviride* isolated from mountain-cultivated ginseng (*Panax ginseng* C.A. Mey.) showed anti-fungal activity against *Botrytis cinerea*, *Pythium* spp., *Rhizoctonia solani*, and *Cylindrocarpon destructans* [70]. The species also enabled a reduction in infection rate of black root rot and stimulated growth and resistance to saline conditions and *Rhizoctonia solani* infection in strawberry (*Fragaria* × *ananassa*) [71]. The ameliorative effect of *T. citrinoviride* on maize growth under salt stress was attributed to its efficient role in the photosynthesis mechanism and osmolyte accumulation [72]. 

The contribution of EPF in increasing the concentration of health-promoting secondary compounds has been a neglected field of research. Most focus has been on the production of medicines by leaf-inhabiting endophytic fungi [73]. Studies on the production of secondary compounds by plants by root-inhabiting EPF are scarce. Verma et al. [74] summarized our current knowledge and referred to the increases in anti-oxidant activity by plants. Our study demonstrated that species of EPF could enhance both plant performance and the production of secondary compounds, although there was a large variability within species of EPF. The closely related species *T. harzianum* and *T. zelobreve* were notably different. While both guilds of fungi (AMF plus EPF) promoted rice performance, their combined activity was additive at best and often showed antagonism (negative synergy), indicating that competition between both guilds prevented the plant from fully benefitting from both groups of fungal symbionts. This (mildly) negative synergy contrasts with the true synergy that has been reported between shoot endophytes of grasses and AMF [75]; although, also in this case, antagonism has been reported under low nitrogen availability [76]. Vignale et al. [77] demonstrated that *Epichloë* (endophytic fungi) exudates promoted colonization by *R. intraradices*, as shown by the higher formation of arbuscules; however, a meta-analysis by Zhong et al. [78] provides a less clear picture. True synergy between AMF and EPF has been rarely reported. Wężowicz et al. [79] reported negative effects by several EPF on the performance of *Verbascum lychnitis* L. and positive effects by AMF, whereas the combination of AMF and EPF had even larger beneficial effects. Khaekhum et al. [25] reported that the growth of sunchoke (*Helianthus tuberosus* L.) was larger when the plants were co-inoculated with EPF *Exserohilum rostratum* and AMF *Claroideoglomus etunicatum* than when inoculated with one fungus only. Unfortunately, they did not statistically test for the interaction term, so it is unclear whether this constitutes a genuine case of synergy. A study by Xu et al. [80] on the combined effects of EPF *Phomopsis liquidambaris* and AMF on the roots of peanuts (*Arachis hypogaea* L.) provided strong suggestions for synergy, but the design of the study did not allow for statistical testing of synergy. 

A major cause for antagonism between AMF and EPF could be due to competition for space in the root. Competition did not result in competitive exclusion as in all cases we found both AMF and EPF in treatments where both fungi were co-inoculated. Even in the absence of EPF, colonization by AMF was low (23.7%). These data are in the range of other studies on rice, for instance, [16] where colonization of 8–14% was reported and [44] with colonization of 29–35%. In our study, competition for resources that the plant provides to both guilds of symbionts was most evident for root colonization (Table 1). The table showed that colonization by AMF drastically reduced colonization by three out of four species of EPF; only in the case of *T. zelobreve* was no reduction observed. The same three species of EPF, when present, also drastically reduced colonization by AMF. Again, *T. zelobreve* was the exception as the combination showed almost complete compatibility. Incompatibility between EPF *T. harzianum* and AMF has been demonstrated earlier by Green et al. [28], and so the full compatibility of the congeneric *Tricoderma* and AMF is somewhat surprising. Other studies indicated that AMF *R. irregularis* suppressed colonization by *T. viride*, but that colonization by AMF remained unaffected by the presence of EPF. Lalaymia et al. [81] noted that a filtrate extract of *T. viride* did not negatively impact colonization by AMF *Rhizophagus irregularis* on the roots of potatoes (*Solanum tuberosum* L.); however, there was no study on how colonization of the roots by *T. viride* impacts root colonization by AMF. 

Competition on the root for colonization by AMF and EPF could then translate into reduced effectiveness. Our study on the combination of AMF and *T. harzianum* showed a fair grain yield and a reasonable production of health-promoting compounds. Dual inoculation of two AMF and *T. asperellum* reduced colonization of AMF on the roots of cacao (*Theobroma cacao* L.) compared with the treatment without EPF, and the combination of AMF and EPF resulted in smaller plants with a higher P mass fraction and a higher disease index of black pod compared to either species individually [82]. Matrood & Rhouma [83] also reported that the combination of *T. harzianum* and AMF *Funneliformis mosseae* was less effective than EPF or AMF when applied separately in reducing Alternaria leaf spot disease in melons (*Cucumis melo* L.). However, their description of the growth medium of *F. mosseae* on sterilized wheat seeds raises doubts about the treatment. The variability of outcomes in the interaction between AMF and different *Trichoderma* species is poorly understood, and the importance of inoculum potential and possible priority effects in colonization demand further attention. 

This study is the first to show successful application of the co-inoculation of AMF *R. variabilis* and EPF *T. zelobreve* on growth promotion and the phytochemical concentrations of Maled Phai rice cultivar, and the effects were far better than the use of mineral fertilizer and non-inoculated plants (control) under pot conditions. Therefore, the development of a bio-fertilizer containing both EPF and AMF for use in rice fields is worth further study. Selections of fungi from both guilds should be based on complementarity or the lack of antagonistic responses in root colonization [84]. Testing under field conditions for the benefits and risks of combinations of AMF and *T. zelobreve* is therefore mandatory.

## 5. Conclusions

This study investigated the potential benefits of co-inoculation of a species of AMF, *R. variabilis,* and several species of EPF. Among these species, the combined inoculation of AMF with *T. zelobreve* enhanced almost all plant growth parameters at the harvesting stage. The combination also did not show competition for space on the roots as neither fungal species reduced root colonization of the other species. These beneficial effects of co- inoculation on plant performance, grain yield, and the production of health-promoting compounds were comparable to or higher than the application of mineral fertilizer. Dual inoculation may be a promising strategy to both reduce expensive synthetic fertilizers, to enhance rice production, and to increase the concentration of health-promoting substances. However, antagonism between AMF and EPF could result in low colonization of both fungal species and as a consequence lead to poor plant performance and lowproduction of health-promoting substances. Testing for the compatibility of fungal inoculants is therefore an essential step for subsequent experimentation. In order to verify these effects, field trials should be undertaken as the next step before these plant growth-promoting fungi can be applied by farmers for the sustainable production of Maled Phai cultivar.

## Figures and Tables

**Figure 1 jof-09-01152-f001:**
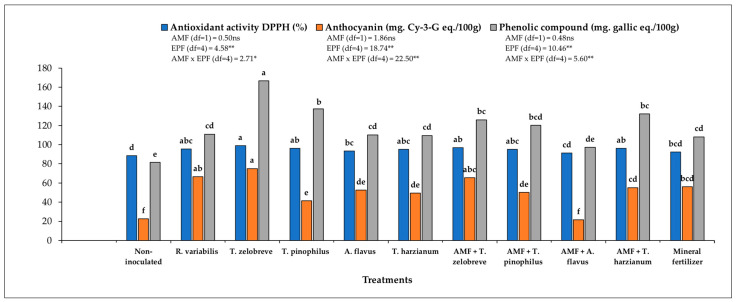
Concentration of anthocyanin and total phenolic compounds and antioxidant capacity of Maled Phai seeds. Different letters indicate significant difference at *p* ≤ 0.05 by LSD. Two-way ANOVA on plant data to test for synergistic effects between AMF and EPF. *, ** and ns: significant at *p* < 0.05, *p* < 0.01 and non-significant probability levels, respectively.

**Table 1 jof-09-01152-t001:** Fractional AMF root colonization and fractional EPF root colonization at harvesting stage.

Treatments	AMF Colonization (%)	EPF Colonization (%)
T1: Non-inoculated (Control)	0.0 d	0 f
T2: *R. variabilis* (AMF)	23.7 a	1 f
T3: *T. zelobreve*	0.0 d	73 ab
T4: *T. pinophilus*	0.1 d	48 d
T5: *A. flavus*	0.0 d	56 c
T6: *T. harzianum*	0.1 d	69 b
T7: AMF + *T. zelobreve*	18.3 b	79 a
T8: AMF + *T. pinophilus*	3.2 c	39 e
T9: AMF + *A. flavus*	2.5 c	42 de
T10: AMF + *T. harzianum*	2.5 c	38 e
T11: Mineral fertilizer	0.0 d	5 f
% CV	15	11
F-test	**	**

Numbers followed by the same letter in each column are not significantly different according to LSD test. **, Significant difference at *p* ≤ 0.01.

**Table 2 jof-09-01152-t002:** Effects of AMF and EPF on growth and production of Maled Phai at harvesting stage.

Treatments	Height (cm)	Shoot Biomass (g)	Root Biomass (g)	Root Diameter (mm)	Panicle Number	Grain Yield (g)
T1: Non-inoculated (Control)	98 f	30.7 d	9.0 ef	0.36 ab	7.8 bcd	7.2 d
T2: *R. variabilis* (AMF)	104 bcd	42.7 bc	17.2 a	0.35 abc	9.5 ab	16.5 a
T3: *T. zelobreve*	104 bc	51.6 ab	11.6 c–f	0.36 a	11.0 a	15.5 ab
T4: *T. pinophilus*	106 b	46.6 ab	9.7 def	0.33 b-e	8.3 bc	13.3 ab
T5: *A. flavus*	99 ef	35.5 cd	8.5 f	0.32 cde	7.8 bcd	12.3 c
T6: *T. harzianum*	110 a	46.5 ab	9.9 def	0.34 a-d	7.3 bcd	14.8 ab
T7: AMF + *T. zelobreve*	112 a	55.4 a	13.5 bc	0.31 e	11.5 a	16.7 a
T8: AMF + *T. pinophilus*	101 cde	45.5 b	10.6 c-f	0.33 b-e	6.5 cd	9.3 d
T9: AMF + *A. flavus*	104 bc	44.5 bc	12.7 cd	0.31 e	5.8 d	7.4 d
T10: AMF + *T. harzianum*	102 b-e	42.3 bc	12.1 cde	0.31 de	8.0 bcd	12.7 bc
T11: Mineral fertilizer	100 def	48.0 ab	16.0 ab	0.32 cde	9.3 ab	14.3 abc
% CV	2	15	19	7	20	15
F-test	**	**	**	**	**	**
	Two-way ANOVA (F-value)					
AMF (df = 1)	2.63 ns	3.72 ns	49.42 **	8.76 **	0.07 ns	0.01 ns
EPF (df = 4)	10.11 **	7.81 **	6.76 **	3.36 *	7.86 **	10.90 **
AMF × EPF (df = 4)	11.40 **	2.18 ns	4.70 **	1.91 ns	1.99 ns	18.49 **

Numbers followed by the same letter in each column are not significantly different according to LSD test. *, Significant difference at *p* ≤ 0.05; **, Significant difference at *p* ≤ 0.01. Two-way ANOVA on plant data to test for synergistic effects between AMF and EPF. *, ** and ns: significant at *p* < 0.05, *p* < 0.01 and non-significant probability levels, respectively.

**Table 3 jof-09-01152-t003:** Effects of AMF and EPF on plant nutrient content of Maled Phai at harvesting stage.

Treatments	Nitrogen (g/Plant)	Phosphorus (g/Plant)	Potassium (g/Plant)
T1: Non-inoculated (Control)	0.178 d	0.028 d	0.300 d
T2: *R. variabilis* (AMF)	0.268 bc	0.033 cd	0.423 bc
T3: *T. zelobreve*	0.385 a	0.038 a–d	0.510 ab
T4: *T. pinophilus*	0.355 a	0.043 abc	0.453 bc
T5: *A. flavus*	0.250 cd	0.038 a-d	0.395 cd
T6: *T. harzianum*	0.328 ab	0.040 a-d	0.488 abc
T7: AMF + *T. zelobreve*	0.360 a	0.048 ab	0.568 a
T8: AMF + *T. pinophilus*	0.368 a	0.048 ab	0.500 ab
T9: AMF + *A. flavus*	0.373 a	0.050 a	0.468 bc
T10: AMF + *T. harzianum*	0.348 a	0.045 abc	0.445 bc
T11: Mineral fertilizer	0.253 c	0.035 bcd	0.453 bc
% CV	16	22	15
F-test	**	*	**
Two-way ANOVA (F-value)			
AMF (df = 1)	7.64 **	6.43 *	5.62 *
EPF (df = 4)	21.32 **	3.40 *	7.18 *
AMF × EPF (df = 4)	2.88 *	0.29 ns	1.52 ns

Numbers followed by the same letter in each column were not significantly different according to LSD test. *, Significant difference at *p* ≤ 0.05; **, Significant difference at *p* ≤ 0.01. Two-way ANOVA on plant data to test for synergistic effects between AMF and EPF. *, ** and ns: significant at *p* < 0.05, *p* < 0.01 and non-significant probability levels, respectively.

**Table 4 jof-09-01152-t004:** Effects of AMF and EPF on chlorophyll content and photosynthesis-related characters of Maled Phai at harvesting stage.

Treatments	Chl a (mg g^−1^ DW)	Chl b (mg g^−1^ DW)	Total Chl (mg g^−1^ DW)	P_n_ (µmol CO_2_ m^−2^ s^−1^)	g_s_ (mol H_2_O m^−2^ s^−1^)	T_r_ (mmol H_2_O m^−2^ s^−1^)	WUE (µmol CO_2_ mmol H_2_O^−1^)
T1: Non-inoculated (Control)	7.2 f	5.6 f	12.7 d	18.0 def	0.28 e	4.4 ef	4.13 bc
T2: *R. variabilis* (AMF)	8.8 bc	16.1 b	24.8 b	19.6 abc	0.43 bc	4.4 ef	4.49 ab
T3: *T. zelobreve*	8.4 cd	12.3 d	20.7 c	19.7 ab	0.49 ab	4.2 f	4.66 a
T4: *T. pinophilus*	7.9 de	5.9 f	13.7 d	18.6 bcd	0.40 cd	5.1 bc	3.70 def
T5: *A. flavus*	5.0 h	8.9 e	13.9 d	16.7 f	0.38 cd	4.9 bc	3.46 ef
T6: *T. harzianum*	4.8 h	8.9 e	13.7 d	18.3 cde	0.34 de	4.7 cde	3.89 cd
T7: AMF + *T. zelobreve*	10.1 a	16.2 b	26.2 b	20.6 a	0.52 a	4.4 def	4.64 a
T8: AMF + *T. pinophilus*	7.6 ef	14.3 c	21.9 c	19.4 abc	0.27 e	5.7 a	3.43 f
T9: AMF + *A. flavus*	6.5 g	14.3 c	20.8 c	17.3 def	0.37 cd	4.4 ef	3.92 cd
T10: AMF + *T. harzianum*	9.2 b	12.3 d	21.5 c	19.7 abc	0.41 cd	5.1 b	3.87 cde
T11: Mineral fertilizer	10.7 a	18.9 a	29.5 a	17.2 ef	0.29 e	4.8 bcd	3.56 def
% CV	6	6	5	5	14	6	7
F-test	**	**	**	**	**	**	**
Two-way ANOVA (F-value)							
AMF (df = 1)	161.35 **	800.93 **	675.00 **	10.19 **	1.45 ns	3.32 ns	0.14 ns
EPF (df = 4)	67.84 **	43.93 **	54.35 **	9.55 **	14.19 **	18.38 **	17.92 **
AMF × EPF (df = 4)	28.09 **	37.96 **	12.64 **	0.35 ns	8.46 **	3.86 *	1.97 ns

Numbers followed by the same letter in each column were not significantly different according to LSD test. *, Significant difference at *p* ≤ 0.05; **, Significant difference at *p* ≤ 0.01. Two-way ANOVA on plant data to test for synergistic effects between AMF and EPF. *, ** and ns: significant at *p* < 0.05, *p* < 0.01 and non-significant probability levels, respectively. Chlo a: Chlorophyll a, Chlo b: Chlorophyll b, Total chlo: Total chlorophyll, Pn: Photosynthesis rate, g_s_: Stomatal conductance, T_r_: Transpiration rate, WUE: Water use efficiency.

## Data Availability

Data are contained within the article.

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
