# Peer review of "Accumulation of Health-Promoting Compounds in Upland Black Rice by Interacting Mycorrhizal and Endophytic Fungi"

_jof, 2023, doi:10.3390/jof9121152_

Round 1

Reviewer 1 Report

Comments and Suggestions for Authors

This research focus on the synergistic effects of an arbuscular mycorrhizal fungus and endophytic fungi on enhancing accumulation of flavonoids, phenolic compounds and antioxidant capacity of upland black rice. It is well known that arbuscular mycorrhizal fungi (AMF) can establish mutualistic associations with the roots of most terrestrial plants, that has a greater ability to acquire nutrients and soil water, and then result in increased plant fitness. Endophytic fungi (EPF), which live inside plant tissues or organs, also can enhance plant growth, improve host tolerance of adverse environmental conditions, enhance the capability of the host plant, or reducing disease severity. This research claimed that the inoculation of AMF had a beneficial effect on black rice growth and yield. What’s more, Co-inoculated AMF (Rhizophagus variabilis KS-02) and EPF (Trichoderma citrinoviride PBMP16) had significantly greater beneficial effects on almost all aspects of plant growth, photosynthesis-related parameters, and yield. They also promoted root growth quality, and plant nutrient uptake.

    However, the phenolic compounds, anthocyanin, and antioxidant capacity in rice seeds harvested from plants co-inoculated with AMF and EPF were dramatically increased compared with those from non-inoculated plants, but the highest content of phenolic and anthocyanin, as well as antioxidant capacity were happened to inoculation of T. citrinoviride. Therefore, I am afraid these results may not make sense in this report.

     Additionally, the content of phenolic, anthocyanin, and antioxidant capacity were tested in seeds. Why not show them in plants? Is there any relationship between the fungi and antioxidant?

    Also, I prefer showing these results in figures instead of tables.

    Additionally, I think it is better to carry out an SEM analysis to show the relationship among the fungal colonization, growth, nutrient content, and chlorophyl and photosynthesis-related characters.

  For the title, it is too long to show the idea clearly.

  For the abstract, it is lack of the key question what the paper focus on.

  For the table 2, it shows that root diameter was the thinnest in the inoculation of R. variabili and T. citrinoviride, but the root mass is very high.

  Lines 43, why was the purple rice showed here, and I do not find any message in following text.

  Line 59 the reference 6 showed be cited in the end of this sentence.

  Lines 100-105 the importance and significance of the research on black rice should be clarified in these sentences.

  Line 192 how do you make sure that the hyphae with septa were beneficial fungi, but not pathogenic fungi.

  Lines 224, what is the sample? How do these samples prepare?

Author Response

Dear Reviewers,

We are grateful to the reviewers who provided constructive criticisms on our manuscript. We have proofing and made changes article in some point based on your suggestions. We are giving the details at each point of editing in this attached document as well.

In addition, we would like to inform the reviewers that EPF Trichoderma citrinoviride as showing in the MS have been changed to the Trichoderma zelobreve which is a new species of Trichoderma in the Harzianum clade from northern China   (Gu et al. 2020). During under review process of the journal, our research group (Gateta et al. 2023) has been re-identified this EPF based on their morphological characteristics and multigene (ITS, rpb2, tef-1, CaM, and BenA) phylogenetic analyses, and the result showed as Tricoderma zelobreve. We changed all place where showing T. citrinoviride to T. zelobreve.

Best regards,

Dr. Nacoon

Response to Reviewer 1

This research focus on the synergistic effects of an arbuscular mycorrhizal fungus and endophytic fungi on enhancing accumulation of flavonoids, phenolic compounds and antioxidant capacity of upland black rice. It is well known that arbuscular mycorrhizal fungi (AMF) can establish mutualistic associations with the roots of most terrestrial plants, that has a greater ability to acquire nutrients and soil water, and then result in increased plant fitness. Endophytic fungi (EPF), which live inside plant tissues or organs, also can enhance plant growth, improve host tolerance of adverse environmental conditions, enhance the capability of the host plant, or reducing disease severity. This research claimed that the inoculation of AMF had a beneficial effect on black rice growth and yield. What’s more, Co-inoculated AMF (Rhizophagus variabilis KS-02) and EPF (Trichoderma citrinoviride PBMP16) had significantly greater beneficial effects on almost all aspects of plant growth, photosynthesis-related parameters, and yield. They also promoted root growth quality, and plant nutrient uptake.

Point 1: However, the phenolic compounds, anthocyanin, and antioxidant capacity in rice seeds harvested from plants co-inoculated with AMF and EPF were dramatically increased compared with those from non-inoculated plants, but the highest content of phenolic and anthocyanin, as well as antioxidant capacity were happened to inoculation of T. citrinoviride. Therefore, I am afraid these results may not make sense in this report.

Response 1: We are unsure about the problem that Reviewer notes. First of all, the word ‘Therefore’ suggests that a conclusion is drawn based on the observations listed. However, there I no logical fallacy here. Responses to both AMF and EPF in terms of plant performance (biomass, nutrient uptake) and in terms of secondary compounds (that may also act as defence compounds) would not necessarily show a perfect correlation. In our view, this differential response (mycorrhiza as significant main factor in nutrient uptake and photosynthetic performance, but not as significant factor for secondary compounds) indicates multiple pathways through fungi influence plant performance (by increasing uptake, by hormonal changes, by upregulating defence compounds, by causing programmed cell death, &c), and it is not surprising that AMF and EPF show differential responses, considering their different evolutionary histories. We have therefore not made changes in the text.

Point 2: Additionally, the content of phenolic, anthocyanin, and antioxidant capacity were tested in seeds. Why not show them in plants? Is there any relationship between the fungi and antioxidant?

Response 2: We investigated those concentrations in rice seeds, as the focus of our research was how to use beneficial fungi to increase health-promoting compound sin human food; and we happen to eat rice seed rather than rice leaves or stems.

Point 3: Also, I prefer showing these results in figures instead of tables.

Response 3: We are changing table 5 to Figure 1.

Point 4: Additionally, I think it is better to carry out an SEM analysis to show the relationship among the fungal colonization, growth, nutrient content, and chlorophyl and photosynthesis-related characters.

Response 4: Thank you for the suggestion. However, there are limits to structural equation modeling, and with 11 treatments (or 10, in case one omits the treatment with mineral fertilizer) and 4 replicates, one is easily limited in the pathways that one can evaluate. This becomes especially problematical if one assumes there are different pathways through which AMF and EPF influence nutrient uptake, photosynthesis, grain production and contents of secondary compounds. Moreover, SEM cannot deal with feedbacks or mutual causation, which creates a problem for a pathway that assesses the effect of AMF on EPF and reversely, where AMF modify colonization of EPF and EPF modify colonization of AMF, as shown in Table 1. This issue of mutual competition is extensively discussed in l. 447-475. The only solution would be to introduce a fairly large number of simplified SEM models and select the model that would fit best. However, that practice is a form of data dredging that should be strongly discouraged, as it provides post-hic explanations rather than contributes to hypothesis testing.

Point 5: For the title, it is too long to show the idea clearly.

Response 5: We changed the title into: Accumulation of health-promoting compounds in upland black rice by interacting mycorrhizal and endophytic fungi

Point 6: For the abstract, it is lack of the key question what the paper focus on.

Response 6: This is listed in l. 24-26. However, to increase clarity we now rephrased that sentence as: “Our main research question was to evaluate the additive and synergistic effects of an arbuscular mycorrhizal fungi and four root-endophytic fungi on plant performance and on the accumulation of health-promoting secondary compounds.”

Point 7: For the table 2, it shows that root diameter was the thinnest in the inoculation of R. variabili and T. citrinoviride, but the root mass is very high.

Response 7: Yes, it has been reported that mycorrhizal fungi (and the same likely pertains to some species of endophytic fungi) modify root architecture by changing cell division rates, initiation of laterals, etc. These could result in changes in root diameter. Fitness of the arbuscular mycorrhizal and endophytic fungus is furthermore enhanced with increasing root biomass, and there are reports of AMF increasing root biomass (there are also reports of AMF resulting in a lower relative root biomass. As these results are what we measured, we did not modify the text.

Point 8: Lines 43, why was the purple rice showed here, and I do not find any message in following text.

Response 8: Black rice is the name commonly used in Thailand. In Europe the name purple rice is frequently used as the rice, when cooked, turns purple because of the anthocyanins. Both terms are equivalent, and we assumed that researchers, interested in these health-promoting rice varieties, know that the are equivalent. However, for clarification we modified the sentence which now reads “Black rice (also known as purple rice)……”

Point 9: Line 59 the reference 6 showed be cited in the end of this sentence.

Response 9: We were not aware that our way of citing literature went against the instructions of Journal of Fungi.

Point 10: Lines 100-105 the importance and significance of the research on black rice should be clarified in these sentences.

Response 10: We added the following lines: “Considering increasing demands for black rice in global markets, it is important that productivity of black rice per hectare increases. Currently productivity of black rice is relatively low, and this constrains its profitability for resource-poor farmers. Productivity increases would be possible by the application of mineral fertilizers, but these can be costly and hence reduce profitability. Harnessing beneficial soil biota such as AMF or endophytic fungi for productivity increases is likely a viable and sustainable alternative under such conditions.”

Point 11: Line 192 how do you make sure that the hyphae with septa were beneficial fungi, but not pathogenic fungi.

Response 11: Our method, recognition of hyphae as septate or not, can indeed not distinguish between septate hyphae of beneficial or antagonistic endophytic fungi. However, as our plants were healthy, and hyphal abundance greatly increased after inoculation with the selected fungi, we assumed that most fungi were the intended EPF. We did therefore not modify the text.

Point 12: Lines 224, what is the sample? How do these samples prepare?

Response 12: a sample is a replicate of a specific treatment; we would think that that is clear to the reader of the paper. Further details of sample preparation can be found in [39], and considering the length of the ms, we decided not to copy that information.

Reviewer 2 Report

Comments and Suggestions for Authors

Overview and general recommendation

The manuscript authored by Nacoon et al. investigated the effects of arbuscular mycorrhizal fungus (AMF) and endophytic fungi (EPF) on enhancing accumulation of anthocyanin, phenolic compounds and antioxidant capacity of upland black rice (Maled Phai cultivar). In brief, the results showed that co-inoculated AMF (Rhizophagus variabilis KS-02) and EPF (Trichoderma citrinoviride PBMP16) had greater beneficial effects on almost all aspects of plant growth, photosynthesis-related parameters, and yield. Meanwhile, the phenolic compounds, anthocyanin, and antioxidant capacity in rice seeds harvested from plants co-inoculated with AMF and EPF were dramatically increased compared with those from non-inoculated plants. 

Comments:

1.      L31: “Trichoderma citrinoviride PBMP16” do you mean with EPF, Trichoderma citrinoviride PBMP16?

2.      The mathematical data is missing in the abstract, please provide some data to support your conclusion.

3.      L113: why Rice seedlings were grown in a greenhouse at 3035 ºC? is it the optimum temperature for rice growth?

4.      L162: what is NPK 15-15-15 mean?

5.      For the results section, could you provide some methermetical data when you compare with the different treatments? I only see the AMF+EPF treatment showed the best perfermence for rice growth, but I cannot see any data provided how much nutrents, phenolic compounds, anthocyanin, and antioxidant capacity in rice increased compared with the other treatment. The authors should be modified the results section and provide the data clearly.

6.      L325-326: if tit is the significant, please provide the p value.

7.      L392: “[58,59,60,61,62,45,63]” is it ok to whrite as [58-62, 45, 63]?

8.      Can you provide some figures to supprot your results, but not all tables? In addition, the AMF and EPM colonization figures also need to be provided.

Comments on the Quality of English Language

english need to check carefully.

Author Response

Dear Reviewers,

We are grateful to the reviewers who provided constructive criticisms on our manuscript. We have proofing and made changes article in some point based on your suggestions. We are giving the details at each point of editing in this attached document as well.

In addition, we would like to inform the reviewers that EPF Trichoderma citrinoviride as showing in the MS have been changed to the Trichoderma zelobreve which is a new species of Trichoderma in the Harzianum clade from northern China   (Gu et al. 2020). During under review process of the journal, our research group (Gateta et al. 2023) has been re-identified this EPF based on their morphological characteristics and multigene (ITS, rpb2, tef-1, CaM, and BenA) phylogenetic analyses, and the result showed as Tricoderma zelobreve. We changed all place where showing T. citrinoviride to T. zelobreve.

Best regards,

Dr. Nacoon

Response to Reviewer 2

The manuscript authored by Nacoon et al. investigated the effects of arbuscular mycorrhizal fungus (AMF) and endophytic fungi (EPF) on enhancing accumulation of anthocyanin, phenolic compounds and antioxidant capacity of upland black rice (Maled Phai cultivar). In brief, the results showed that co-inoculated AMF (Rhizophagus variabilis KS-02) and EPF (Trichoderma citrinoviride PBMP16) had greater beneficial effects on almost all aspects of plant growth, photosynthesis-related parameters, and yield. Meanwhile, the phenolic compounds, anthocyanin, and antioxidant capacity in rice seeds harvested from plants co-inoculated with AMF and EPF were dramatically increased compared with those from non-inoculated plants. 

Comments:

L31: “Trichoderma citrinoviride PBMP16” do you mean with EPF, Trichoderma citrinoviride PBMP16?

Response: Yes, one of the EPF used. (showing as T. zelobreve in the MS).

The mathematical data is missing in the abstract, please provide some data to support your conclusion.

Response: We are unsure what the reviewer exactly wants. We refer to significant effects, with automatically imply p < 0.05. Giving numbers for plant biomass, nutrient content, photosynthetic performance, and concentrations of secondar compounds would make the ms much lengthier than is advised by the journal. If readers are interested in the topic and find the Abstract inviting, they can easily see from tables and Figures the exact results.

L113: why Rice seedlings were grown in a greenhouse at 30−35 ºC? is it the optimum temperature for rice growth?

Response: Yes, it is

L162: what is NPK 15-15-15 mean?

Response: NPK 15:15:15 refers to a mixture of mineral fertilizers, containing N, P and k at 15% weight basis

For the results section, could you provide some met hermetical data when you compare with the different treatments? I only see the AMF+EPF treatment showed the best performance for rice growth, but I cannot see any data provided how much nutrients, phenolic compounds, anthocyanin, and antioxidant capacity in rice increased compared with the other treatment. The authors should be modified the results section and provide the data clearly.

Response: These data are in the various tables, where it shown which main factor and / or interaction terms are significant; and whether individual treatment differ significantly. Adding that an increase from 98 to 104 cm (plant height in table 2, comparing T1 and T2) constitute an increase of 4% does, in our view, not add substantially to the results. For Table 1 (which gives already percentages) fractional change would possibly be even confusing.

L325-326: if tit is the significant, please provide the p value.

Response: The general EPF effect is listed in table 4 as 0.01<p<0.05; different letters in the table indicate significant differences at p < 0.05. What additional data are asked for?

L392: “[58,59,60,61,62,45,63]” is it ok to write as [58-62, 45, 63]?

Response: Thank you; we now modified as [45, 58-63], as that is more convenient.

Can you provide some figures to support your results, but not all tables? In addition, the AMF and EPM colonization figures also need to be provided.

Response: While it is a matter of taste what one prefers (some people love numbers in tables, as that allows doing additional calculations; while other people are more visually oriented and prefer graphs), here we revised version has table 5 replaced by figures 1. This pertains f.i. for table 1 (colonization by AMF and EPF, which has not become a figure).

Reviewer 3 Report

Comments and Suggestions for Authors

Authors presented research on additive and synergistic effects of AMF and EPF on the growth and some physiological parameters of black rice. The experimental set up, methodology, results and discussion are all very well presented. I have no major concerns, I can only congratulate authors for the very good and clearly presented paper. 

Some minor details, maybe to consider:

1. In the Table 1, it is visible that AMF colonization is rather low. Could you include somewhere in the Discussion are these values of about 20% usual for the black rice?

2. Also about very low colonization of black rice with AMF when accompanied with EPF - is it possible to discuss could such low percentage of AMF colonization (2-3%) mean anything at all for the plant host in other studies.

3. Concusions - I suggest to repeat that besides the conclusion that T. citrinoviride + R. variabilis combination is quite efficient for the black rice and quite promisinig for the application, other EPF in combination with R. variabilis gave variable results, sometimes even indicating negative sinergy. So the selection of appropriate combinations of beneficial microorganisms is obviously very important. It does not function in every case.

Author Response

Dear Reviewers,

We are grateful to the reviewers who provided constructive criticisms on our manuscript. We have proofing and made changes article in some point based on your suggestions. We are giving the details at each point of editing in this attached document as well.

In addition, we would like to inform the reviewers that EPF Trichoderma citrinoviride as showing in the MS have been changed to the Trichoderma zelobreve which is a new species of Trichoderma in the Harzianum clade from northern China   (Gu et al. 2020). During under review process of the journal, our research group (Gateta et al. 2023) has been re-identified this EPF based on their morphological characteristics and multigene (ITS, rpb2, tef-1, CaM, and BenA) phylogenetic analyses, and the result showed as Tricoderma zelobreve. We changed all place where showing T. citrinoviride to T. zelobreve.

Best regards,

Dr. Nacoon

Response to Reviewer 3

Authors presented research on additive and synergistic effects of AMF and EPF on the growth and some physiological parameters of black rice. The experimental set up, methodology, results and discussion are all very well presented. I have no major concerns, I can only congratulate authors for the very good and clearly presented paper. 

Some minor details, maybe to consider:

  1. In the Table 1, it is visible that AMF colonization is rather low. Could you include somewhere in the Discussion are these values of about 20% usual for the black rice?

Response: We added in l. 448, after the first sentence of that paragraph: “Competition did not result in competitive exclusion as in all cases we found both the AMF and the EPF in treatments where both fungi were co-inoculated. Even in the absence of EPF colonization by AMF was low (23.7%). These data are in the range of other studies on rice, for instance [16], where colonization of 8-14% was reported and [45] with colonization 29-35%.

  1. Also about very low colonization of black rice with AMF when accompanied with EPF - is it possible to discuss could such low percentage of AMF colonization (2-3%) mean anything at all for the plant host in other studies.

Response: We refer to the strong negative effect when combining AMF with some (not all) EPF and attribute this to competition. However, while these cases also resulted in poorer plant performance, we cannot attribute the effects to low colonization per se or the negative effects of EPF. One would need time-course studies to assess how rapidly antagonistic interactions between both fungal guilds occur and what initial effects would then be shown, but that question was beyond the scope of our study.

  1. Concusions - I suggest repeating that besides the conclusion that T. citrinoviride + R. variabilis combination is quite efficient for the black rice and quite promisinig for the application, other EPF in combination with R. variabilis gave variable results, sometimes even indicating negative sinergy. So the selection of appropriate combinations of beneficial microorganisms is obviously very important. It does not function in every case.

Response: Thank you for your suggestion. We added, after l. 496. “However, antagonism between AMF and EDF could result in low colonization of both fungal species and as a consequence lead to poor plant performance and low production of health-promoting substances. Testing for compatibility of fungal inoculants is therefore an essential step for subsequent experimentation.”